# Genomic Features of MCR-1 and Extended-Spectrum β-Lactamase-Producing Enterobacterales from Retail Raw Chicken in Egypt

**DOI:** 10.3390/microorganisms9010195

**Published:** 2021-01-19

**Authors:** Mustafa Sadek, José Manuel Ortiz de la Rosa, Mohamed Abdelfattah Maky, Mohamed Korashe Dandrawy, Patrice Nordmann, Laurent Poirel

**Affiliations:** 1Medical and Molecular Microbiology, Department of Medicine, Faculty of Science and Medicine, University of Fribourg, CH-1700 Fribourg, Switzerland; mustafa.sadek@unifr.ch (M.S.); jose.ortizdelarosa@unifr.ch (J.M.O.d.l.R.); patrice.nordmann@unifr.ch (P.N.); 2INSERM European Unit (IAME, France), University of Fribourg, CH-1700 Fribourg, Switzerland; 3Department of Food Hygiene and Control, Faculty of Veterinary Medicine, South Valley University, Qena 83522, Egypt; mohamed.maky@gmail.com (M.A.M.); mohamed.dandrawy@gmail.com (M.K.D.); 4Swiss National Reference Center for Emerging Antibiotic Resistance (NARA), University of Fribourg, CH-1700 Fribourg, Switzerland; 5Institute for Microbiology, University of Lausanne and University Hospital Centre, CH-1011 Lausanne, Switzerland

**Keywords:** plasmid-mediated colistin resistance, MCR-1, ESBLs, *fosA4*, poultry

## Abstract

Colistin is considered as a last resort agent for treatment of severe infections caused by carbapenem-resistant Enterobacterales (CRE). Recently, plasmid-mediated colistin resistance genes (*mcr* type) have been reported, mainly corresponding to *mcr-1* producers. Those *mcr-1*-positive Enterobacterales have been identified not only from human isolates, but also from food samples, from animal specimens and from environmental samples in various parts of the world. Our study focused on the occurrence and characterization of *mcr-1*-positive Enterobacterales recovered from retail raw chicken in Egypt. From the 345 retail chicken carcasses collected, a total of 20 samples allowed to recover *mcr-1*-positive isolates (*Escherichia coli*, *n* = 19; *Citrobacter freundii*, *n* = 1). No *mcr-2-* to *mcr-10*-positive isolate was identified from those samples. The colistin resistance trait was confirmed for all those 20 isolates with a positivity of the Rapid Polymyxin NP (Nordmann-Poirel) test. Minimum inhibitory concentrations (MICs) of colistin for all MCR-1-producing isolates ranged between 4 and 16 μg/mL. Noticeably, 9 out of the 20 *mcr-1*-positive isolates produced an extended-spectrum β-lactamase (ESBL), respectively producing CTX-M-9 (*n* = 2), CTX-M-14 (*n* = 4), CTX-M-15 (*n* = 2), and SHV-12 (*n* = 1). Noteworthy, the *fosA4* gene encoding resistance to fosfomycin was found in a single *mcr-1*-positive *E. coli* isolate, in which both genes were located on different conjugative plasmids. The pulsed-field gel electrophoresis (PFGE) patterns were identified, corresponding to 10 different sequence types (STs), highlighting the genetic diversity of those different *E. coli*. Whole-genome sequencing revealed three major types of *mcr-1*-bearing plasmids, corresponding to IncI2, IncX4, and IncHI2 scaffolds. The occurrence of MCR-1-producing multidrug-resistant Enterobacterales in retail raw chicken is of great concern, considering the possibility of transmission to humans through the food chain.

## 1. Introduction

The worldwide increase and spread of the plasmid-mediated colistin resistance (MCR), in particular in multidrug-resistant (MDR) bacteria of human and animal origin, is a major public health concern [1]. Colistin resistance was mainly linked to chromosomal mutation(s) or deletion(s) of the phoPQ and pmrAB two-component systems, involved in the biosynthesis of the lipopolysaccharide [2]. The *mcr-1* transferable resistance gene was first identified on a conjugative IncI2 plasmid in China in 2015 [3]. Since then, nine additional MCR-like encoding genes have been reported, namely *mcr-2* to *10* [4,5,6,7,8,9,10,11,12]. MCR proteins are phosphoethanolamine transferases that modify the lipid A moiety of the lipopolysaccharide of Gram-negative bacteria, leading to decreased susceptibility or resistance to either polymyxin B or colistin [3]. The spread of the *mcr-1* gene is associated with various plasmid backbones such as IncI2, IncX4, IncFI, IncFII, IncFIB, IncHI1, IncHI2, IncP, and IncY, the two first being the most prevalent [3,13,14,15,16,17,18].

The *mcr-1* gene has been reported mostly from Enterobacterales recovered from food, animals, humans, and the environment in various parts of the world [19]. The extensive use of colistin in veterinary medicine for prophylactic and therapeutic purposes as well as for growth promotion has been recognized as a major risk factor for the emergence and dissemination of colistin-resistant *Escherichia coli* isolates in food-producing animals [19,20]. Hence, the food chain may significantly increase the dissemination and acquisition of colistin-resistance worldwide.

The *mcr-1*-carrying plasmids in *E. coli* identified from human and from poultry meat in Switzerland were highly similar, indicating that certain types of epidemic plasmids such as IncI2 and IncX4 play an important role in the dissemination of the *mcr-1* gene along the food chain and in humans [20]. In Africa, *mcr-1*-positive *E. coli* have been detected in food items, including chicken meat and chicken carcasses [21,22], sausage [23], cheese [24], and raw milk [25]. However, there is still a lack of comprehensive data concerning the epidemiology of *mcr* genes in *E. coli* of food origin in African countries, including Egypt [26]. This study reports the isolation and characterization of foodborne *E. coli* strains carrying *mcr-1* gene, *fosA*, and extended-spectrum ß-lactamase (ESBL)-encoding genes recovered from retail raw chicken in Egypt.

## 2. Materials and Methods

### 2.1. Bacterial Isolates, Susceptibility Testing

Between July and December 2018, a total of 345 retail chicken carcasses were randomly collected from different poultry slaughterhouses, supermarkets, and butcher shops in different Egyptian cities (Qena, Luxor, Nag Hammadi, and Esna city). The neck skin samples (25 g) taken from each carcass were homogenized and enriched in buffer peptone water (225 mL) for 24 h at 37 °C with shaking. Colistin-resistant isolates were recovered by direct spreading on MacConkey agar supplemented with 2 µg/mL colistin. Colonies of different morphology, size, and color from each plate were selected and used for further analysis, and subsequently submitted to a screening of *mcr* genes. For that purpose, DNA was extracted from all colistin-resistant strains with the QIAamp DNA mini-kit and the QIAcube workstation (Qiagen, Courtaboeuf, France), according to the manufacturer’s instructions. Then, PCR screening for plasmid-mediated colistin resistance genes (*mcr-1* to *10*) was performed [12,27]. *mcr*-positive isolates were subsequently selected for further characterization and analysis. Antimicrobial susceptibility testing was performed by the disk diffusion method according to the Clinical and Laboratory Standards Institute (CLSI) recommendations (CLSI M100 ED30:2020), on Muller-Hinton agar plates broad-spectrum cephalosporins, carbapenems, aminoglycosides, aztreonam, quinolones, sulfonamides, tigecycline, and fosofmycin. Minimum inhibitory concentrations (MICs) of colistin were evaluated by broth microdilution in cation-adjusted Mueller-Hinton broth (Bio-Rad, Cressier, Switzerland), as recommended by the European Committee on Antimicrobial Susceptibility Testing (EUCAST)/Clinical & Laboratory Standards Institute (CLSI) joint guidelines (https://eucast.org/fileadmin/src/media/PDFs/EUCAST_files/General_documents/Recommendations_for_MIC_determination_of_colistin_March_2016.pdf). Results were interpreted according to the European Committee on Antimicrobial Susceptibility Testing (EUCAST)/CLSI joint guidelines (www.eucast.org). The MIC of fosfomycin was determined by the agar dilution method using cation-adjusted Mueller-Hinton agar (MHA-CA, reference 64884; Bio-Rad, Marnes-la-Coquette, France) supplemented with glucose-6-phosphate (25 μg/mL), as recommended by CLSI, 2020 (CLSI M100 ED30:2020).

### 2.2. Phenotypic Confirmation of Resistance Patterns

Extended-Spectrum β-Lactamase (ESBL)-producing and colistin-resistant isolates were tested by the Rapid ESBL NP (Nordmann-Poirel) test and the Rapid Polymyxin NP test respectively [28,29], in order to confirm their resistance patterns. The Rapid Fosfomycin NP test was performed as described [30]. For all selected isolates, identification at the species level was performed using the API20E system (bioMérieux, La-Balme-les-Grottes, France).

### 2.3. Molecular Assays for Other Resistance Determinants

The identification of ESBL-encoding genes (namely *bla*_TEM_, *bla*_SHV_, and *bla*_CTX-M_) was performed by PCR using previously reported primers [31]. Additionally, PCR amplification was performed to detect any known plasmid-mediated *fosA* genes (*fosA1* to *fosA8*), as previously described [32]. All positive PCR amplicons were sent for sequencing (Microsynth, Balgach, Switzerland).

### 2.4. Mating-Out Assays

The transferability of all *mcr-1*-positive isolates was investigated by a filter-mating assay, as described previously [33], in which *mcr-1*-positive isolates were used as donors and azide-resistant *E. coli* J53 as the recipient. Transconjugants were selected on LB agar supplemented with sodium azide (100 µg/mL) and colistin (2 µg/mL). Transconjugants were obtained for all donors at 25–30 °C or 37 °C. Transconjugants were confirmed by PCR targeting the *mcr* genes, and antimicrobial susceptibility testing.

### 2.5. Plasmid Isolation, Plasmid Analysis

The plasmids carrying the *mcr-1* gene from *E. coli* transconjugants were typed by using the PCR-based replicon typing (PBRT) method, as previously reported [34]. The PCR scheme was complemented with primers specific for the IncX3-type plasmids [35], as well as primers specific for IncX4 plasmids [36]. The size of the plasmid was obtained after Kieser extraction for the resulting *E. coli* transconjugant strains [37], followed by gel electrophoresis analysis, with reference strain *E. coli* 50,192 containing 4 plasmids (154, 66, 48, and 7 kb, respectively) being used as a molecular marker.

### 2.6. Clonality Evaluation

The clonal relationship of *E. coli* isolates was evaluated by pulsed-field gel electrophoresis (PFGE) analysis, as described previously [35,38], and multi-locus sequence typing (MLST). Seven housekeeping genes were used for *E. coli* (*adk, fumC, gyrB, icd, mdh, purA,* and *recA*). MLST analyses were performed according to EnteroBase (http://mlst.warwick.ac.uk/mlst/dbs/*Ecoli*). For PFGE, total DNA from *E. coli* isolates was digested using the *XbaI* enzyme (New England BioLabs, Ipswich, USA). Then, the generated fragments were separated by PFGE using a CHEF-DR III System (Bio-Rad, Cressier, Switzerland).

### 2.7. Plasmid Sequencing and Bioinformatic Analysis

Seven representative *mcr-1*-plasmids were selected for sequencing by using Illumina technology. Templates used corresponded to plasmid extracts obtained for the *E. coli* transconjugants using the Qiagen Large Plasmid Construct kit (Qiagen, Hilden, Germany). Genomic libraries were assessed using a Nextera XT library preparation kit (Illumina Inc., San Diego, CA, USA), and sequencing was performed using an Illumina MiniSeq system with 150 bp paired-end reads. De-novo genome assembly was performed using the CLC Genomic Workbench (version 20.0.4; CLC Bio, Aarhus, Denmark), and contigs with a minimum contig length of 800 nt were generated. The resulting assembled sequences were uploaded to the Center for Genomic Epidemiology server (http://www.genomicepidemiology.org/). The MLST and plasmid replicon types were determined using the MLST (version 2.0.4) and PlasmidFinder (version 2.0.1) software, respectively. Antimicrobial resistance was analyzed by ResFinder 4.1 (https://cge.cbs.dtu.dk/services/ResFinder/) and CARD (https://card.mcmaster.ca/analyze/rgi). The constructed plasmids were annotated automatically by the RAST server using the RAST-tk scheme [39], followed by manual inspection and correction using the BLASTn and BLASTp tools (https://blast.ncbi.nlm.nih.gov/Blast.cgi). The circular image and comparisons between other reported similar plasmids were performed using the BLAST Ring Image Generator (BRIG) tool (Figure 1) [40].

## 3. Results and Discussion

**Occurrence of MCR-1 producers****:** From the 345 retail chicken carcasses, a total of twenty samples allowed to recover *mcr-1*-positive isolates (*E. coli*, *n* = 19; *Citrobacter freundii*, *n* = 1). No *mcr-2-* to *mcr-10*-positive isolate was identified from those samples. The colistin resistance trait was confirmed for those twenty isolates by the positivity of the Rapid Polymyxin NP test. Determination of the MICs of colistin for all MCR-1-producing isolates showed that they ranged from 4 to 16 μg/mL (Table 1). Although the current situation of antimicrobial use for livestock is not precisely known in Egypt, colistin usage is common in animal husbandry in Egyptian farms, including for calves, poultry (for example, treatment of colibacillosis), and rabbits [41,42]. Therefore, it is tempting to speculate that such selection of colistin-resistant *E. coli* isolates among Egyptian farms might be related to the overuse of colistin in the local farming industry [41]. Two *mcr-1*-positive *E. coli* strains isolated from humans have been identified in Egypt so far [43]. The *mcr-1* gene had also been identified from *E. coli* isolates recovered from a diseased cow [44], and from healthy broilers [45]. Concerning strains isolated from food, a single *mcr-1*-positive *E. coli* strain was isolated from cheese [24] and beef sausage [23] in Egypt.

**Susceptibility patterns, resistance mechanisms, and clonal relationship:** Noticeably, among these twenty *mcr-1*-positive isolates, nine isolates showed an extended-spectrum β-lactamase (ESBL) phenotype that was related to the presence of the following genes, namely *bla*_CTX-M-9_ (*n* = 2), *bla*_CTX-M-14_ (*n* = 4), *bla*_CTX-M-15_ (*n* = 2), and *bla*_SHV-12_ (*n* = 1). A recent study demonstrated that *mcr-1* occurred more frequently among ESBL-producing *E. coli* than among non-ESBL-producing *E. coli* [46]. The occurrence of the *mcr-1* gene was found to be high among ESBL-producing *E. coli* recovered from broiler chickens that died from colibacillosis in Tunisia [22]. In other studies, the co-occurrence of ESBL and MCR-1 have been reported from different origins [16,47,48,49,50,51]. It has been shown that the ESBL and *mcr-1* genes could be co-localized on the same conjugative plasmid [15,46,52,53,54,55].

All but four MCR-1-producing isolates identified in this study possessed the *bla*_TEM-1_ gene (Table 1). All these MCR-1-producing isolates showed resistance to chloramphenicol and sulfonamides. A majority also exhibited resistance to amoxicillin (90%), tetracycline (95%), cephalothin, sulfamethoxazole/trimethoprim, ciprofloxacin, and gentamicin (50%), and kanamycin (60%). It is noteworthy that the high rates of resistance observed for sulfonamides, chloramphenicol, and tetracycline correlate with the extensive usage of those antibiotics in veterinary medicine [2]. Among the nineteen MCR-1-producing *E. coli* isolates identified in this study, ten different sequence types were identified, namely ST101, ST156, ST371, ST373, ST398, ST986, ST1011, ST1125, ST1196, and ST5687, highlighting the genetic diversity of those MCR-1 positive *E. coli* isolates (Table 1).

**Identification of FosA4 fosfomycin resistance determinant:** It is noteworthy that, among these twenty *mcr-1*-positive isolates identified in this study, a single isolate was found to be fosfomycin-resistant (positive Rapid Fosfomycin NP test, MIC of fosfomycin, >256 mg/L) that was shown to be associated to the presence of the *fosA4* gene, which had not been co-transferred with *mcr-1*. In that isolate, the *mcr-1* and *fosA4* genes were located on IncX4 and IncFII plasmids, respectively. The *fosA4* gene was previously reported in the literature from human clinical *E. coli* isolates in Japan and Australia [56,57]. The identification of FosA determinants remains rare among animal isolates, with few reports so far, corresponding to the occurrence of the *fosA3* gene among *E. coli* isolates from different animal sources (cattle, pigs, poultry, and pet animals) in China, Brazil, and France [58,59,60,61,62]. To our knowledge, our study reports the first *fosA4*-positive *E. coli* isolate from an animal source.

**Plasmid characterization:** Conjugative assays and PCR-based replicon typing (PBRT) analysis were performed for all *mcr-1*-positive isolates. Transconjugants were obtained for all donors, indicating that the *mcr-1* gene was located on self-transferable conjugative plasmids (Table 1). Those *E. coli* transconjugants exhibited MIC values for colistin at 4 or 8 µg/mL, while that of the *E. coli* J53 recipient strain was found at 0.25 µg/mL. No additional resistance trait was transferred along with the *mcr-1* gene in any of the *E. coli* transconjugants, except for EC65.2 TC that showed resistance to tetracycline, chloramphenicol, and sulfonamides. PBRT analysis revealed that the *mcr-1* gene was localized on different plasmid scaffolds differing in sizes and structures, including IncI2 (55%), IncX4 (30%), and IncHI2 (15%) (Table 1). Those findings are consistent with previous reports that the worldwide spread of the *mcr-1* gene is mainly driven by three major plasmid types—IncI2, IncX4, and IncHI2 [1,19,63,64,65,66,67,68]—with IncI2 being the most prevalent plasmid backbone, followed by IncX4 and IncHI2.

Considering that several plasmid types were identified, seven representative *mcr-1*-positive plasmids were entirely sequenced. WGS data analysis showed that the seven plasmids belonged to three types of plasmids, including IncX4 of size ranging from 31.8 to 32.1 kb (*n* = 3), IncI2 of size ranging 64.2–67.7 kb (*n* = 3), and IncHI2 of 195.8 kb in size (*n* = 1). None of the *mcr-1*-positive IncI2 and IncX4 plasmids possessed additional resistance determinants, confirming the phenotypic observations made with the *E. coli* transconjugants, whereas multiple resistance elements were detected alongside *mcr-1* in the pEGYMCR65 plasmid. The three sequenced IncX4 type plasmids (pEGYMCR8, pEGYMCR16, pEGYMCR60) were nearly identical, and showed typical plasmid backbones encompassing genes encoding proteins involved in replication, maintenance, and transfer.

Interestingly, detailed sequence analysis of the IncX4 plasmids showed that they were almost identical to the *mcr-1*-positive plasmid pCFSAN061769_01 (~97% query coverage and ~99.9% sequence identity; GenBank accession no. CP042970.1) identified in an *E. coli* recovered from cheese in Egypt, indicating that this plasmid type is circulating among different sources. The sequence of those IncI2 plasmids (pEGYMCR17, pEGYMCR23, pEGYMCR62) was almost identical to plasmid pEGYMCR-1, identified in a single *E. coli* isolate recovered from a meat product (beef sausage) in Egypt, and to plasmid pMCR-GN775 (accession no. KY471307) identified in an *E. coli* strain recovered from a gastrostomy tube site and rectum of a patient hospitalized in Canada who noticeably received previous healthcare in Egypt (Figure 1) [23,69]. This basically underlines that the dissemination of the *mcr-1* gene is at least partially linked to that of “epidemic” plasmids.

The sequence of plasmid pEGYMCR65 was highly similar to that of the *mcr-1*-positive IncH12 plasmid, pCFS3292-1 (95% query coverage and 99.99% sequence identity; GenBank accession no. CP026936.2), identified from an *E. coli* of bovine origin recovered from animals presenting with diarrhea and mastitis in France. Also, plasmid pEGYMCR65 was highly similar to the *mcr-1*-positive plasmid pEGY1-MCR-1 (91% query coverage and 99.99% sequence identity; NCBI Reference Sequence: NZ_CP023143.1) identified in *E. coli* isolated from cheese in Egypt (Figure 1), that latter missing transposon Tn*21* encoding mercury resistance genes (*mer* operon).

**Genetic environment of the *mcr-1* gene:** In a recent study, an intermediate circular form of the insertion sequence IS*Apl1* associated with *mcr-1* was detected, suggesting that IS*Apl1* might play a driving role in the horizontal gene transfer of this resistance gene [70]. IS*Apl1* (usually only one copy) is found upstream of the *mcr-1* gene. Recently, it has been reported that a second copy of IS*Apl1* may be found downstream of the *mcr-1* gene, therefore forming the composite transposon Tn*6330* [71,72]. By further analysis of the genetic context surrounding the *mcr-1* genes identified in this study, the *mcr-1*-*pap2* element was identified on all plasmids. The intact composite transposon Tn*6330* (IS*ApI1*-*mcr-1*-*orf*-IS*ApI1*) was identified on IncHI2 plasmids. IncX4 and IncI2 plasmids did not display an IS*Apl1* element in the vicinity of the *mcr-1* gene (Figure 1). Our results are consistent with previous reports showing that circular intermediates could be detected mainly in IncHI2 plasmids, not in IncX4 or IncI2 plasmids [23,54,69,70,73].

## 4. Conclusions

Our findings highlight the occurrence of MCR-1- and ESBL-producing Enterobacterale strains in poultry farms in Egypt. It might be speculated that the overuse and misuse of both β-lactams and colistin in veterinary practice might be the main cause of such worrying phenomenon. A high diversity of genetic structures responsible for the acquisition of *mcr-1* in Enterobacterales, and particularly in *E. coli*, was shown here by focusing on those isolates recovered from retail chicken meat from Egypt. The detection of plasmid-mediated colistin resistance in poultry meat is of potential public health concern since it is considered as an important source of transmission of plasmid-mediated *mcr-1* to consumers through handling and eating of raw or undercooked meat. Our findings suggest that the local and global spread of the *mcr-1* gene has mainly been driven by three major plasmid types, with IncI2 being the most prevalent plasmid backbone, followed by IncX4 and IncHI2. Effective intervention approaches aiming to reduce antibiotic selective pressure in food-producing animals in low-income countries including Egypt must absolutely be implemented, to prevent further selection of multidrug-resistant bacteria.

## Figures and Tables

**Figure 1 microorganisms-09-00195-f001:**
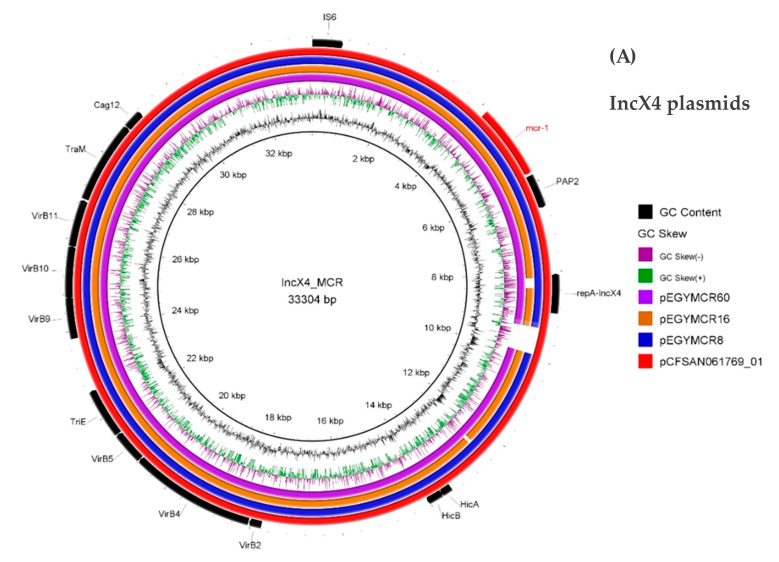
Circular map of *mcr-1*-positive plasmids compared to other reported similar plasmids. Panel (**A**) IncX4 plasmids, panel (**B**) IncI2, panel (**C**) IncHI2. Red outer ring: plasmid used as reference for the alignment; size of the reference indicated in the middle of each panel. The different arrows indicate the positions, directions of transcription, and predicted function of the ORFs. The *mcr-1* gene and IS*Apl1* are marked in red and blue, respectively. The circular maps were generated using the BRIG tool.

**Table 1 microorganisms-09-00195-t001:** Phenotypic and genetic features associated with MCR-1-producing *E. coli* isolates from chicken carcasses, Egypt.

Strains	Species	Origin (Source)	ST (CC)	Resistance Genes	MIC of Colistin (µg/mL)	Rapid Polymyxin NP Test	Rapid ESBL NP Test	Resistance Profile	*mcr-1*-Harboring Plasmid		
									Inc Group (Kb)	MIC Colistin (mg/L)	Co-Resistance Markers
EC10.2	*E. coli*	Chicken carcass	ST101 (CC101)	*mcr-1*, *bla*_TEM-1_	4	+	-	AMX, TIC, SUL, NAL, CIP, CHL, TET, SXT, GMI, TMN	IncI2 (≈66)	4	none
EC16.1	*E. coli*	Chicken carcass	ST1196	*mcr-1*, *bla*_TEM-1_, *bla*_CTX-M-9_	4	+	+	AMX, TIC, PIL, CTX, CEF, FEP, TET, TMN, SXT, KMN, CIP, GMI, NAL, SUL, CHL	IncX4 (31.8)	4	none
EC20.1	*E. coli*	Chicken carcass	ST1196	*mcr-1*, *bla*_TEM-1_, *bla*_CTX-M-9_	8	+	+	AMX, TIC, PIL, FEP, CTX, CEF, NAL, CIP, TET, SUL, CHL, KMN, GMI, TMN, SXT	IncX4 (≈32)	4	none
EC8.1	*E. coli*	Chicken carcass	ST371 (CC350)	*mcr-1*, *fosA*4, *bla*_TEM-1_	4	+	-	AMX, TIC, PIL, SUL, NAL, KMN, TET, SXT, FOS	IncX4 (32.1)	4	none
EC23.1	*E. coli*	Chicken carcass	ST398 (CC398)	*mcr-1*, *bla*_TEM-1_, *bla*_SHV-12_	4	+	+	AMX, TIC, PIL, CEF, CZD, ATM, CTX, NAL, CIP, TET, SUL, CHL, SXT, KMN, GMI, TMN	IncI2 (67.7)	4	none
EC49.2	*E. coli*	Chicken carcass	ST1125	*mcr-1*, *bla*_TEM-1_	4	+	-	AMX, TIC, PIL, SUL, CHL, TET	IncI2 (≈66)	4	none
EC62.2	*E. coli*	Chicken carcass	ST5687	*mcr-1*, *bla*_CTX-M-15_	4	+	+	AMX, TIC, PIL, CEF, FEP, CZD, ATM, CTX, NAL, CIP, TET, SUL, CHL, KMN, GMI, SXT	IncI2 (64.5)	4	none
EC59.1	*E. coli*	Chicken carcass	ST5687	*mcr-1*, *bla*_CTX-M-15_	4	+	+	AMX, TIC, PIL, FEP, CEF, CZD, ATM, CTX, NAL, CIP, TET, SUL, CHL, KMN, GMI, SXT	IncI2 (≈66)	4	none
EC65.2	*E. coli*	Chicken carcass	ST1011	*mcr-1*, *bla*_TEM-1_, *bla*_CTX-M-14_	16	+	+	AMX, TIC, PIL, CTX, FEP, CEF, NAL, CIP, TET, SUL, CHL, KMN, GMI, SXT.	IncHI2 (195.8)	8	TET, SUL, CHL
EC53.2	*E. coli*	Chicken carcass	ST156 (CC156)	*mcr-1*, *bla*_TEM-1_, *bla*_CTX-M-14_	4	+	+	AMX, TIC, PIL, CEF, CTX, NAL, CIP, TET, SUL, CHL, KMN, GMI, TMN	IncX4 (≈32)	4	none
EC60.2	*E. coli*	Chicken carcass	ST156 (CC156)	*mcr-1*, *bla*_TEM-1_, *bla*_CTX-M-14_	4	+	+	AMX, TIC, PIL, CEF, CTX, NAL, CIP, TET, SUL, CHL, KMN, GMI, TMN	IncX4 (32.1)	4	none
EC54.2	*E. coli*	Chicken carcass	ST156 (CC156)	*mcr-1*, *bla*_TEM-1_, *bla*_CTX-M-14_	8	+	+	AMX, PIL, TIC, CEF, CTX, NAL, CIP, TET, SUL, CHL, KMN, TMN, GMI	IncX4 (≈32)	4	none
EC56.2	*E. coli*	Chicken carcass	ST986	*mcr-1*	8	+	-	TET, SUL, CHL, KMN	IncHI2 (>154)	8	TET, SUL, CHL, KMN
EC57.2	*E. coli*	Chicken carcass	ST986	*mcr-1*	8	+	-	TET, SUL, CHL, KMN	IncHI2 (>154)	4	TET, SUL, CHL, KMN
EC52.2	*E. coli*	Chicken carcass	Unknown	*mcr-1*, *bla*_TEM-1_	8	+	-	AMX, TIC, SUL, CHL, SXT, TET	IncI2 (≈66)	8	none
CF12.2	*C. freundii*	Chicken carcass	ND	*mcr-1*, *bla*_TEM-1_	8	+	-	AMX, TIC, FOX, CEF, AMC, CHL, SUL	IncI2 (≈66)	8	none
EC17.1	*E. coli*	Chicken carcass	ST373 (CC168)	*mcr-1*, *bla*_TEM-1_	8	+	-	AMX, TIC, SUL, CHL, TET, PIL, SXT, NAL	IncI2 (64.2)	4	none
EC18.2	*E. coli*	Chicken carcass	Unknown	*mcr-1*, *bla*_TEM-1_	8	+	-	AMX, TIC, PIL, AMC, SUL, CHL, NAL, TET, FOX	IncI2 (≈66)	4	none
EC15.2	*E. coli*	Chicken carcass	ST373 (CC168)	*mcr-1*, *bla*_TEM-1_	8	+	-	AMX, TIC, SUL, CHL, TET	IncI2 (≈66)	8	none
EC13.1	*E. coli*	Chicken carcass	ST373 (CC168)	*mcr-1*, *bla*_TEM-1_	8	+	-	AMX, TIC, PIL, TET, SUL, NAL, CHL	IncI2 (≈66)	4	none

Abbreviations: CC, clonal complex; Inc., plasmid incompatibility group; ST, sequence type; MIC, minimal inhibitory concentration; AMX, amoxicillin; TIC, ticarcillin; PIL, piperacillin; CTX, cefotaxime; CZD, ceftazidime; ATM, aztreonam; CEF, cephalothin; FEP, cefepime; TET, tetracycline; TMN, tobramycin; SXT, sulfamethoxazole/trimethoprim; KMN, kanamycin; CIP, ciprofloxacin; GMI, gentamicin; NAL, nalidixic acid; SUL, sulfonamides; CHL, chloramphenicol; ND, not determined; +, resistant; -, susceptible.

## Data Availability

The complete nucleotide sequences of pEGYMCR_IncHI2, pEGYMCR_IncI2, and pEGYMCR_IncX4 were deposited as GenBank accession numbers MT499884, MT499885, and MT499886, respectively.

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
