# Peer review of "Genomic Features of MCR-1 and Extended-Spectrum β-Lactamase-Producing Enterobacterales from Retail Raw Chicken in Egypt"

_microorganisms, 2021, doi:10.3390/microorganisms9010195_

Round 1
Reviewer 1 Report
This study focused on the occurrence and characterization of mcr-1-positive Enterobacterales recovered from the retail raw chicken. Moreover it highlights the need to reduce the antibiotic selective pressure in food-producing animals in middle-income countries like Egypt to prevent further selection of multidrug-resistant bacteria. The detection of plasmid-mediated colistin resistance in poultry meat is of potential public health concern since it is considered as an important source of transmission of plasmid-mediated mcr-1 to consumers.
The topic is interesting and well written. The chapters are well constructed and the experimental methods are clear, innovative, and understandable. Results and conclusions are consistent. A high diversity of genetic structures responsible for the acquisition of mcr-1 in E. coli was shown by authors. The bibliography is consistent, adequate and updated.
The paper should be considered for publication in its present form.
Best regards
Author Response
Many thanks for the reviewer for this positive feedback to our submitted manuscript.
Reviewer 2 Report
The authors investigated relevant aspects in terms of phenotypic and genotypic antibiotic resistance profiles in foods of animal origin. The research was developed by recent investigation techniques and an appropriate scientific design. The epidemiological data obtained are relevant. The article is very well written by consulting an appropriate bibliography. My compliments.
The study deserves to be published.
Author Response
Many thanks for the reviewer for this positive feedback to our submitted manuscript. For the English language and style, the whole manuscript was revised very thoroughly.
Reviewer 3 Report
The authors have studied the occurrence and characterization of mcr-1-positive Enterobacterales recovered from retail raw chicken in Egypt, founding 20 positives samples.
Minor amendments included in the file attached. The authors must clarify the table and the figures.

Author Response
All the comments were amended as suggested by the reviewer.
Reviewer 4 Report
The manuscript is well written and carefully prepared, but I would like to address a few minor comments:
Abstract: line 22: the species name „Escherichia coli” was used for the first time, therefore the full name should be used
Line 65: Authors must specify: are they mentioned genes or proteins? and correct the form of the spelling
Line 71: Please explain why this type of sample was selected for the study
Line 83 and others: Please provide the CLSI Standard Release Year that was used in the study. Include the standard used in the reference list
Line 95: Have the Authors analyzed the presence qAmpC? Occasionally, both ESBL and AmpC can coexist in the same strain. Please give a few words of explanation
Line 201: The word "noteworthy" repeated in the same paragraph, please correct it
Line 270: Please correct the typo
Author Response
The manuscript is well written and carefully prepared, but I would like to address a few minor comments:
Abstract: line 22: the species name „Escherichia coli” was used for the first time, therefore the full name should be used
1. Done as suggested by the reviewer.
Line 65: Authors must specify: are they mentioned genes or proteins? and correct the form of the spelling
2. Genes. Corrected as suggested by the reviewer.
Line 71: Please explain why this type of sample was selected for the study
3. The neck skin is preferred sampling location because it is the most contaminated part of poultry carcass containing a high bacterial load owing to the slaughter technology at which the poultry hangs up site down with the neck skin to the bottom. Moreover, the removal of neck skin for microbiological examination did not decrease the quality grade of the chicken carcass, as would the removal of breast skin.
Line 83 and others: Please provide the CLSI Standard Release Year that was used in the study. Include the standard used in the reference list
4. The reference was provided as suggested by the reviewer.
Line 95: Have the Authors analyzed the presence qAmpC? Occasionally, both ESBL and AmpC can coexist in the same strain. Please give a few words of explanation
5. Among these mcr-1-positive isolates, 9 isolates showed an extended-spectrum β-lactamase (ESBL) phenotype using disc diffusion and the Rapid ESBL NP test. After that, those 9 isolates were analyzed by PCR for the presence of ESBL encoding genes but weren’t analyzed for AmpC.
Line 201: The word "noteworthy" repeated in the same paragraph, please correct it
6. Corrected as suggested by the reviewer.
Line 270: Please correct the typo
7. It is “Enterobacterale” but it is because of the journal format.